# How Can We Improve the Consumption of a Nutritionally Balanced Maternal Diet in Rural Bangladesh? The Key Elements of the “Balanced Plate” Intervention

**DOI:** 10.3390/ijerph17176289

**Published:** 2020-08-28

**Authors:** Ashraful Alam, Morseda Chowdhury, Michael J. Dibley, Camille Raynes-Greenow

**Affiliations:** 1School of Public Health, The University of Sydney, Sydney 2006, Australia; michael.dibley@sydney.edu.au (M.J.D.); Camille.raynes-greenow@sydney.edu.au (C.R.-G.); 2Health, Nutrition and Population Programme, Bangladesh Rural Advancement Committee (BRAC), Dhaka 1212, Bangladesh; morseda.c@brac.net

**Keywords:** maternal diet, dietary behaviour change intervention, nutrition education, balanced plate, process evaluation, qualitative methods, Bangladesh

## Abstract

Social, cultural, environmental and economic factors closely regulate the selection, allocation and consumption of maternal diets. We developed a nutrition behaviour change intervention to promote a balanced diet in pregnancy through practical demonstration in rural Bangladesh and tested the impact with a cluster randomised controlled trial. This paper presents the findings of the process evaluation and describes the strategies that worked for intervention compliance. We conducted in-depth interviews with pregnant women, women who birthed recently, and their husbands; focus groups with mothers and mothers-in-law; key-informant interviews with community health workers, and observations of home visits. We identified six key areas within the intervention strategy that played a crucial role in achieving the desired adherence. These included practical demonstration of portion sizes; addressing local food perceptions; demystifying animal-source foods; engaging husbands and mothers-in-law; leveraging women’s social networks; and harnessing community health workers’ social role. Practical demonstration, opportunity to participate and convenience of making of the plate with the food available in their kitchen or neighbours’ kitchen were the most commonly mentioned reasons for acceptance of the intervention by the women and their families. The balanced plate intervention helped women through practical demonstration to learn about a balanced meal by highlighting appropriate portion sizes and food diversity. The women needed active involvement of community health workers in mobilising social support to create an enabling environment essential to bring changes in dietary behaviours. Future implementation of the intervention should tailor the strategies to the local context to ensure optimal adherence to the intervention.

## 1. Introduction

Most of the 211 million women who become pregnant each year live in developing countries [1]. Despite many efforts, more than 10 million children and 529,000 mothers die mainly from avoidable causes [1]. Maternal and child undernutrition indirectly contributes to 3.5 million deaths and 35% of the disease burden in children younger than five years and 11% of the total global disability-adjusted life years (DALYs) [2].

Maternal undernutrition with chronic energy and micronutrient deficiencies are most prevalent in south-central Asia. Nutritional deficiency in pre-pregnancy and pregnancy is associated with adverse outcomes such as low birth weight (LBW), preterm birth, and intrauterine growth restriction (IUGR). All these adverse outcomes have an influence on survival, development, quality of life, and health care costs [3].

Undernutrition usually associates with inadequate dietary intake but can also be linked to increased nutritional requirements, nutritional losses, and reduced ability to absorb or utilise nutrients [4]. Inadequate intake can result from a lack of access and availability of food, known as food insecurity, or the improper utilisation in the quality and quantity and inequitable intra-household distribution of food [5]. Social, cultural, environmental and economic factors tightly regulate the selection, allocation and consumption of food [6].

In Asia, Latin America and Africa, a wide range of restrictions on maternal dietary practices exist on the basis of perceptions around food being beneficial or damaging [7]. External influences, such as advice and social judgment, can potentially influence eating choices. Pregnancy is a transitional period that offers a window of opportunity for women to change behaviour towards the adoption of a healthier diet that can improve the health and well-being of the mother and her baby [8]. In a systematic review, nutrition education provided during pregnancy was effective in improving dietary behaviours in pregnant women [9] including in pregnant adolescents [10] and resulting in increased consumption of protein and specific vitamins, and improved birthweight [11,12]. However, the general quality of the evidence of the studies included was considered low due to methodological weaknesses. Hence, the systematic review advocated for new research that was well-designed and grounded in appropriate theories of behaviour change to improve the evidence of the effect of nutrition education and counselling on pregnancy outcomes [12].

An interdisciplinary team developed a nutrition behaviour change intervention with a goal of promoting a balanced diet in pregnancy and increasing the birthweight of neonates in rural Bangladesh, and evaluated it in a cluster randomised controlled trial (cRCT) [13]. During the final months of the intervention, the authors of this paper conducted a qualitative process evaluation to assess how the participants viewed the intervention and the elements of it that played a crucial role in achieving the desired adherence. This paper presents the findings and insights obtained in the process evaluation. The impact assessments will be documented in a separate paper.

### Cluster Randomised Controlled Trial

To provide with the context of this process evaluation study, we will concisely describe the cluster randomised controlled trial (cRCT) within which we conducted the process evaluation. The trial protocol has been published previously [13]; briefly, a two-arm parallel cRCT was conducted in villages in the Sherpur district in Bangladesh. Thirty-six clusters were randomly selected and assigned to one of two treatment arms. We enrolled twenty-five pregnant women from each cluster; 445 into the balanced plate nutrition education group who received nutrition counselling and practical demonstrations from early pregnancy until childbirth (“balanced plate” intervention) and 448 into the standard nutrition education group (usual care). The cRCT hypothesised that infants would be 100 g heavier in the intervention group rather than the comparison group. Trained community-based health workers (Shasthya Kormis or SKs) with at least secondary education, who were employed by the Bangladesh Rural Advancement Committee also known as BRAC (a large NGO) [14], visited all women once a month to provide antenatal care services. These services included nutrition education (either balanced plate intervention, or the usual standard nutrition education). The balanced plate nutrition education combined counselling on eating a nutritionally balanced meal, food diversity, and a demonstration of making a nutritionally balanced plate (meal) with appropriate portion sizes from foods available in the women’s kitchens. The intervention included any family members, especially those who were decision-makers to become involved to motivate and support their pregnant family member to adopt a healthy diet and female-friendly intra-household food distribution.

A culturally appropriate daily meal menu (Table 1) and a pictorial food chart were produced based on the currently recommended diet for Bangladeshi pregnant women. The menu contained foods from all seven-food groups (cereal, lentil, animal protein, vegetables, fruits, milk and oil) giving 2500 kcal energy daily, and included essential micro and macronutrients. The balanced plate that we used for the demonstration was a combination of foods in appropriate portion size, essential to meet the requirements of a pregnant woman.

The trial was approved by the James P. Grant School of Public Health, BRAC University Ethical Review Committee, Dhaka, Bangladesh. The cRCT was registered with the Australia New Zealand Clinical Trials Registry (ACTRN12616000080426).

## 2. Materials and Methods

### 2.1. Study Population and Sampling

For this process evaluation, we selected a sub-sample of the women who participated in the cRCT. We used a stratified purposeful sampling method [15] to identify eligible participants, women who were pregnant (regardless of gestational age), or women who had birthed within the last three months and had a living child. We invited the husbands, mothers, and mothers-in-law of these pregnant or recently birthed women to participate. Shasthya Kormis were selected randomly from the intervention clusters.

We applied three qualitative methods—in-depth interviewing (IDI), focus group discussion (FGD), and key-informant interviewing (KII) (Table 2). We conducted four FGDs—two with pregnant women, one with husbands, and one with mothers and mothers-in-law. The research included 16 IDIs with women. We held the KIIs with all four Shasthya Kormis who delivered the balanced plate intervention, and we also conducted observations of the home visits.

### 2.2. Data Collection

Semi-structured guidelines guided our data collection. We developed separate guidelines for IDI, FGD and KII, and translated into Bengali. Morseda Chowdhury (MC) designed the guidelines after a literature review, and in consultation with Ashraful Alam) (AA) and a nutritionist. The interview guidelines were pretested (by MC and a local anthropologist) with four pregnant women and two husbands in the study area. Guidelines were revised based on their responses and feedback. We trained the interviewers after the finalisation of the questionnaire and guidelines. Data collection took place during November–December 2016. With support from the local BRAC staff, MC filtered one potential respondent from each registry book of Shasthya Kormis based on the eligibility criteria. Prior to the interview, the interviewer explained the purpose of the study, confidentially of the information and respondents’ right to withdraw, and obtained informed consent. We obtained consent from all participants and the family decision-makers (if present), as well as consent to digitally record the interview. We used coded numbers to maintain the anonymity of the data.

We used IDIs to elicit information about how women and family members made choices of what and how much of the different food groups to eat. The interviewers focused the conversations on four broad topics: (1) opinions on balanced plate counselling, (2) benefits of balanced plate demonstration as a tool of nutrition education, (3) role of husband, mother and mother-in-law in steering food accessibility and intra-household food distribution, (4) barriers and enablers to complying with the balanced plate education messages in daily life. FGDs were conducted to gather views of the family decision-makers on the balanced plate intervention and whether it brought changes in the household budget allocation to food and intra-household food distribution. Discussion topics included benefits and weaknesses of the balanced plate intervention, constraints of accessing food choices and the strategies to overcome constraints, and the role of older generation women in regulating pregnant women’s food share in the household.

To explore the health providers’ perspectives of community acceptance of the balanced plate intervention, we interviewed the Shasthya Kormis as key informants who delivered the intervention. The major discussion points were the role of Shasthya Kormis in promoting pregnant women’s health and nutrition in the community, community perceptions of the balanced plate intervention, and coping strategy of managing additional tasks within regular work schedules.

### 2.3. Data Analysis

We audio-recorded all interviews and discussions. One of the authors (MC) transcribed and translated them into English for analysis. We used both inductive and deductive coding approaches. We used a priori codes to identify the text related to the research questions and added additional codes to the codebook based on new themes that emerged during coding. Initially, MC manually coded one interview and one focus group transcript following the data collection guidelines to develop a draft codebook. A senior qualitative researcher (AA) checked the codebook to improve inter-coder reliability. Subsequently, MC coded all the transcripts. While coding, she maintained flexibility of modifying the codebook by adding any new topics that emerged throughout the coding process. Later, we compiled the text about each thematic code into separate files. These files were read and re-read by MC and analysed, using a thematic approach [16]. We looked for variations, similarities and emerging trends in themes. Iterative discussions within the research team were organised to make a consensus about the themes, understand the issues and interpret the findings.

## 3. Results

All participants reported their participation in the trial favourably and gave a favourable review of the intervention. The most common reasons for the attractiveness of the intervention were the practical demonstration, the opportunity to participate actively, and the convenience of preparing the plate with the foods available in their own or their neighbours’ kitchens. Our qualitative investigation identified six key areas within the intervention strategy that played a crucial role in achieving high adherence.

### 3.1. Practical Demonstration of Portion Sizes Engaging the Users

The women and the family members generally mentioned that they liked the demonstration of making the balanced plate. The most common reason was that this was a new method and different from other nutrition education interventions they had experienced.

Participants found the practical demonstration more attractive because it was participatory and needed active involvement in gathering essential food items, measuring foods and displaying them on a plate. They did not have to buy anything that was used in the plate. All the foods used were from their kitchen, which made them feel comfortable with the process. Moreover, the women considered measuring the food as an innovation that allowed them to see the exact amount of each food item they had to eat. This approach was clearer than their previous advice and knowledge of eating “more” food.

*This is a modern method; I have never seen something like this before. She (Shasthya Kormi) talked about different kinds of nutritious foods, why they are important, and how they will benefit our child. We all participated in the process of measuring foods and then put on a plate. I think this is something new*.—Husband.

*My daughter-in-law does not want to eat, and she doesn’t feel like (eating). Now she would realise how little she is eating. She has to eat that much as shown on the plate. Now she will listen to what I say (for eating more)*.—Mother-in-law.

The pregnant women considered that the visual presentation was effective in assisting their recall of the messages and was easy to apply in practice. For example, one pregnant woman explained that initially, she used the bowl that was shown to measure the food but later developed a new method of measuring similar portion sizes with spoons. She said the display of the foods on a plate left a strong visual impression for her. Subsequently she repeated the same plate from memory without even seeing the menu or using the measuring bowl.

*Rozina Apa (fictitious name of Shasthya Kormi) showed me how to make my plate. She used a bowl from my kitchen and measured rice, curry, lentil and other foods and put together on a plate. First couple of days I also tried that way, but later I realised that I know how much rice makes two bowls, I don’t need to measure it now. I can do it with a spoon*.—Pregnant woman.

*Displaying visually made it easy for me to recall what components should be on the plate. For example, now I know in my plate there should be either meat or fish or egg. I should eat lentil regularly, which I was not habituated before*.—Pregnant woman.

Similar to the food tradition of rice being the dominate portion of a meal in rural Bangladesh, the women in the study used to usually consume a large portion of rice every day with a few other foods. They reported that despite having some knowledge of “good” foods, they did not know about the appropriate portion sizes for different foods before the intervention. They confirmed that the Shasthya Kormis of the project taught them about correct measurement of rice and other foods in each meal.

### 3.2. Addressing Local Food Perceptions

Women and family members perceived fruit as a “good food”, but the price determined the value of the fruit. Most of the women said that they were inclined to eat imported fruits which are more expensive, as these were perceived to be more nutritious than the inexpensive or freely available local fruits. The fruits imported from overseas such as apples, oranges and grapes, which families used to buy on special occasions or when someone in the family was unwell, were considered highly nutritious. From the intervention, the women and family members said, they understood that local fruits like guava, jujube, banana, and carambola were equally good and that pregnant women should eat these seasonal fruits. These fruits were available in the trees owned by them or in the vicinity. They were available in the local market and cost less than apples, oranges or grapes.

It was evident from the interviews and observations that the Shasthya Kormis emphasised changing the perceptions that guided the food selection behaviour. They counselled to make the women aware that the market price did not determine the nutritional value of a food. They advised the women that the inexpensive local fruits are as good as the costly foreign fruits. Shasthya Kormis also discussed that many women preferred sour fruits because they improved their appetite. This information helped them advise several specific local fruits to consume during pregnancy such as jujube, carambola and Burmese grape.

### 3.3. Demystifying Animal-Source Foods

Most of the women mentioned that their husbands liked the idea of eating a balanced diet as promoted by the Shasthya Kormi. Husbands also tried to buy as many of the foods as they could afford. However, the husbands reported that they could only afford expensive foods like beef or chicken once or twice a week or even a month. Part of the intervention was teaching the participants about the availability of low-cost fish, that also nutritious. The participants highly valued this advice. With the perception that meat is one of the most nutritious foods, the information that fish and egg were of similar nutrient value was new to the families. The women usually ate fish quite often but in a small quantity. As a result of the counselling, the families tried to increase the amount of fish in their daily menu. The families opted to buy inexpensive farmed fish such as tilapia, carp and catfishes as opposed to expensive fish varieties as a measure to increase the amount of animal-source food.

*Previously my husband used to buy big fish once in 1–2 months. I know it is expensive and not affordable for him; still, he used to buy, even in small quantity. Now we don’t aim for buying big fish; he buys tilapia, pangash (a variety of white catfish) and the like but more often. You don’t need much money to buy them these days*.—Pregnant woman.

### 3.4. Engaging Husbands and Mothers-in-Law

Participating families reported reallocating the existing budget or earning more to spend on food. Husbands said the counselling and demonstration influenced them and they were convinced that pregnant women should eat more food with a greater diversity. They often changed their shopping patterns to accommodate more healthy food. To do so, they said, they often stretched the budget. Some husbands said they had channelled some expenses that they thought unnecessary (e.g., ready-made snacks) to buy more fish, milk and fruits. Several said that they worked to secure extra earnings to purchase additional food. One husband shared his experience of managing the extra budget required:

*Suppose my previous weekly budget (for food) was taka 1500. Now I need additional taka 200–300 per week. I work some extra hours to manage the extra money my family needs now. I have no problem with it*.—Husband.

Women echoed their husbands, saying that their husbands were buying more of the foods suggested by the Shasthya Kormi and doing it willingly.

*He (husband) bought more foods while I was pregnant. He didn’t wait for me to tell him what to buy*.—Recently delivered woman.

The woman gave mixed responses when asked about the role of their mother-in-law on their diet. Some women said their mothers-in-law did not bother about what they ate, while others mentioned that they told them to eat more. However, the largest share of food always went to the male household heads. One mother-in-law justified allocation of a larger share of food to the man, arguing that male members need to stay fit for work.


*My son is the person who had to work all day long to feed the whole family. Where would he get energy for work if he does not eat more? Can they do it if they don’t eat meat and fish?*


During the intervention visits, the Shasthya Kormis explored family members’ cooperation with the women for their balanced diet consumption. They negotiated with the family member who created an obstacle with consent from the woman. Engaging the mother-in-law helped:

*This is my third child. I feel craving for food all the time. My mother-in-law watched me and didn’t allow me to eat stomach-full. She says the baby will get big. So, I asked Apa (Shasthya Kormi) to talk to her but without letting her know that I told her about it. She might get angry if she comes to know, you know. I found afterwards that my mother-in-law is not preventing me from eating some more*.—Recently delivered woman

### 3.5. Involving Neighbours

During the intervention visits at the beginning of the trial, the Shasthya Kormis welcomed interested women from the neighbouring families. If a household did not have all groups of foods needed to make a balanced plate, neighbours came forward with a contribution. This experience led the women to occasionally leverage their social networks within the neighbourhood to fulfil the shortage of any food items in the plate. Women exchanged foods with their neighbours, borrowed food, and in relatively poorer families, accepted neighbours’ food contributions. The Shasthya Kormis said that they encouraged the women to keep the neighbours involved throughout the trial. The women reported that the neighbours continued their support of them.

*The women around me didn’t receive any benefit (from this project), but (they) helped me so that I can make the plate. It’s natural that we didn’t always have everything (all food items) to make the plate as exactly Apa’s (Shasthya Kormi) has told us. There were some days when I had to exchange foods with my neighbours*.—Pregnant woman.

### 3.6. Harnessing Community Health Workers’ Social Role

Most of the pregnant women said that Shasthya Kormis were easy to reach and could be called for help anytime. The participants held the Shasthya Kormis in high esteem who had a long-term connection with the community.

Interviews with women and focus groups with the mothers-in-law discovered that the Shasthya Kormis played a crucial role in negotiating an equitable distribution of food in favour of pregnant women. The women maintained that, as respectable members of the community, the Shasthya Kormis were accepted by the families in dealing with sensitive issues like intra-household food distribution. The women expressed their concerns freely and confidently to the Shasthya Kormi with a hope for a solution.

*Jahanara Apa (a Shasthya Kormi) has been working in our village for years, and we know her very well. She is my relative too. She is seeing (ANC visit) pregnant women for a long time. So, I am happy to discuss with her, and I am sure she is the best person to help me here*.—Pregnant woman.

Mothers-in-law expressed that Shasthya Kormis’ repeated contact motivated them to change the attitude toward improved and increased diet of their pregnant daughters-in-law.

## 4. Discussion

In this paper, we identified the critical strategies of our intervention including the importance of a practical demonstration of meal portion sizes, social support, family participation in the intervention and consideration of local cultural elements. We discovered that the balanced plate nutrition intervention through a practical demonstration helped women to learn about a balanced meal by considering appropriate portion sizes and dietary diversity. The decision-makers in the family, such as husbands and mothers-in-law, adopted a healthier diet to protect the newborn and the mother from complications necessary to have a healthy baby. We also found that the first level health workers (Shasthya Kormis in the study) played a useful role in mobilising family support to effect change in the dietary habits of the women. We identified the critical elements in the intervention strategy that improved the end-user’s capacity to adhere to the maternal nutrition education messages delivered in our intervention.

Both women and families in the study liked the participatory approach to preparing a balanced plate because the method was new to them and allowed them to engage in the formulation of the meal actively. It indicated that people liked to contribute with their ideas rather than only being instructed to follow some pre-fixed advice. In a society like Bangladesh, where rice provides around 70% of total energy with little animal protein and vegetables [17], it is challenging to alter families’ dietary patterns. However, women in our study modified their diet after they realised the imbalanced food proportions in their usual meals. Previous research also reported evidence of the impact of practical demonstration approach to health education on oral health [18] and breastfeeding promotion [19].

Given the complexity of human dietary behaviour, studies have acknowledged that only processing knowledge of health benefits is not enough for motivating dietary change [20,21,22,23,24]. Evidence from a systematic review suggests the importance of additional factors such as attitudes and social norms to be considered when designing a nutrition education intervention [12]. Addressing these mediating variables is essential to understanding the local food perceptions that guide foods choice and consumption. In Lao PDR, pregnant women increased consumption of the foods they perceived to be “good” in nutrition to fulfil their desire of improved health and growth of their foetus [25]. Our study explored women’s perceptions of nutritious foods and individual food taste to identify the foods to be suggested for inclusion in the balanced plate. This specificity of information made our nutrition education precise to the needs that in turn, become acceptable to the end-users.

High prices compromise the demand and intake of animal-source foods among low-income families. For example, in Kenya, Cornelsen and colleagues reported that price was the most commonly reported reason for not consuming animal-source food [26]. Further aggravating the price effects on demand for animal-source foods is the perception that only high-cost foods have higher nutritional value or quality, which then drives food choice [27,28]. The families accepted our strategy of demystifying animal-source food by providing nutrition information about locally available low-cost fish varieties and changed their food purchasing behaviours.

In low- and middle-income settings, women often do not have self-efficacy regarding their health and nutrition and tend not to prioritise their own needs. Women with less influence or power within the household are unable to guarantee equal food distribution in their families [29]. Mothers-in-law are important gate-keepers of pregnancy and neonatal care behaviours in these settings including South Asia [30,31,32,33]. In China, nutrition education targeted at postpartum women did not bring desired behaviour change because their mothers or mothers-in-law cared for them during the “sitting month” (first 30 or 40 days postpartum) without allowing them the autonomy to decide their care practices including diet. Targeting family members, rather than only women, was recommended by the study to acquire optimum benefits from nutrition education intervention [34].

The social expectation of men often restricts the husbands’ role in this women’s domain to a limited supportive role. Still, their involvement is decisive in the food choices of families in Bangladesh because household expenditure and going to the market are traditionally the domains of men. Evaluation of a randomised trial attributed a 25% increase in maternal dietary diversity to the engagement of husbands in a maternal nutrition program in rural Bangladesh [35]. In our study, mothers-in-law valued pregnancy and the family members desired a good outcome both for mother and baby. Husbands’ engagement in the counselling influenced them to seek opportunities to earn more to buy foods for their wives.

Evidence suggests that changing behaviour requires confidence in one’s ability to behave in a modified pattern of practice [36]. A favourable environment with social support [37], family support [38] and support from others such as health professionals is a pre-requisite to this confidence [39]. Apart from changing maternal dietary behaviour, we found improvement in community support such as neighbours’ contribution to food items to make a balanced plate. Studies showed similar success in infant feeding practices, micronutrient supplementation and dietary diversity through a combined effort of influencing women by health workers, husbands and mothers-in-law to overcome cultural elements and help to develop best practices [35,40].

In an assessment of the implementation of the Bangladesh national nutrition services, there were no usual providers in the rural setting trained in nutritional and dietary advice [41]. Thus, it is likely that pregnant women received very brief nutritional messages from doctors if they received any, which was insufficient to impact dietary behaviour. The “WHO Global Strategy on Human Resources for Health (HRH): Workforce 2030” encourages countries to implement a diverse, sustainable skills mix, harnessing the potential of community health workers [42]. The 2013 Lancet Maternal and Child Nutrition Series also emphasised community health worker-led interventions at a scale to reach more people [43]. In this study, we have demonstrated the decisive role of the community health workers in addressing maternal undernutrition in rural Bangladesh.

Furthermore, beyond the community health workers’ conventional health provider role, in the project we uniquely harnessed the social position of the Shasthya Kormis to create an enabling environment for optimal maternal nutrition. Social scientists have suggested adopting a broader social role for community health workers to achieve better public health [44].

Our intervention went beyond transferring nutrition facts [45] or information dissemination components [46,47]. The intervention’s inclusiveness of motivational, action and environmental elements [46,47] has increased its adherence. We ensured inclusiveness by verbal communication to raise awareness and enhance the motivation of the pregnant woman; practical demonstration to facilitate the ability to act; and engaging the influential family members and neighbours to create an enabling environment.

## 5. Conclusions

We identified the strategies that worked to enhance adherence to the balanced plate nutrition education intervention in the context of the local setting. The findings provide an important message for future interventions for improving intervention uptake. Future implementation of the intervention should tailor the strategies to the local context to ensure optimal adherence to the intervention.

## Figures and Tables

**Table 1 ijerph-17-06289-t001:** The balanced plate menu for pregnant women.

Meal	Food	Quantity (One Dish = 250 mL)
Breakfast	Rice	1.5 dishes
	or	
	Chapati (medium size)	3 pieces
	Vegetables	1 dish
	Egg	1
	or	
	Lentil (thick)	1 dish
Mid-morning snack	Seasonal fruit(s)	1 piece/dish
	Milk product(s)	1 dish
Lunch	Rice	3 dishes
	Lentil (thick)	1 dish
	Leafy/non-leafy vegetables	1.5 dishes
	Meat/fish/egg	1 piece
Afternoon snack	Milk	1 glass
	Seasonal fruit(s)	1 piece/dish
	Puffed rice with molasses	1 dish
	or	1 dish
	Biscuits	
Dinner	Rice	2 dishes
	Lentil (thick)	1 dish
	Leafy/non-leafy vegetables	1.5 dishes
	Meat/fish/egg	1 piece
	Milk	1 glass
	or	
	Curd	0.5 dish

**Table 2 ijerph-17-06289-t002:** Methods and sample size.

Method	Type of Respondents	Number
In-depth interviews	Mothers of infants	16
Focus group discussions	Pregnant women	2
	Mothers-in-law	1
	Husbands	1
Key informant interviews	Shasthya Kormi	4
Observation of Shasthya Kormis’ intervention home visits		13

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
