# Peer review of "How Can We Improve the Consumption of a Nutritionally Balanced Maternal Diet in Rural Bangladesh? The Key Elements of the “Balanced Plate” Intervention"

_ijerph, 2020, doi:10.3390/ijerph17176289_

Round 1
Reviewer 1 Report
This paper aimed to understand the strategies to improve the maternal diet of a group of pregnant women from Bangladesh.
Although the aim of the study is of absolute interest, the results and conclusions of this work would seem to be based solely on comments, experiences and anecdotes reported by a group of pregnant women and their families who participated in a nutrition education intervention. There would seem to be no real measurement of food consumption, only a self-declaration by women that they have changed their diet.
On this basis, I find it difficult to understand some of the authors' statements, for example:
- In the results section, as regard to portions demonstrations, the authors report that women in the study confirmed that they learnt about how much rice and other foods they should eat in each meal. Did you evaluate the food consumption to assess if women really understood the meaning of right food portions?
- In the results section, the authors report that women understood the importance of local products, or low-cost foods such as eggs and fish instead of meat, and that the families tried to increase the consumption of these foods. Was the consumption of these foods measured before the intervention and at the time of the interview? Is the increased/decreased consumption statistically and nutritionally significant? In the absence of this, how can you be sure of the dietary change?
- Similarly, the authors report that some husbands said they changed their shooping patterns to accommodate healthier food. Did you check their food purchases before and after the nutritional intervention or the change is just self-declared by the subject?
Apparently, these assessments were not made during the study.
Other concerns:
- Is there a statistical analysis to support the results and conclusions?
- Also, why were the women who followed the standard intervention not interviewed? Have they maintained the same dietary habits or changed them?
Although I generally agree that a demonstration intervention involving participants is more effective than a traditional passive lesson, in this case I find it difficult to accept the conclusion that "Programs to improve maternal diet quality should focus on promoting a healthy diet through practical demonstration of portion sizes and active engagement of the women and family instead of replicating the traditional information-based counselling" on the basis of the presented results.
Author Response
Thank you for your comments. Below, we have addressed point-by-point the comments.
Comment: Although the aim of the study is of absolute interest, the results and conclusions of this work would seem to be based solely on comments, experiences and anecdotes reported by a group of pregnant women and their families who participated in a nutrition education intervention. There would seem to be no real measurement of food consumption, only a self-declaration by women that they have changed their diet.
Response: We thank the review for the comment. Our paper presents the findings of the qualitative process evaluation of a cluster randomized controlled trial (cRCT) of nutritionally-balanced maternal diet intervention. As described in the abstract and the introduction of the manuscript (lines 77-79), rather than presenting the impact outcome data of the cRCT, this paper disseminates the findings of the process evaluation of the study. Our objective in this paper as stated in the manuscript (lines 78-79) is “To assess how the participants viewed the intervention and the elements of it that played a crucial role in achieving the desired adherence.” In this paper, we focused on describing the strategies that worked for intervention compliance, not the impact outcome measures. However, the impact outcomes data of the intervention trial is the aim of another paper that is under review by a journal. Process evaluation findings make valuable contribution to generating practical knowledge that helps program implementers replicate and scale-up similar programs. Thus, results of process evaluation of intervention are being increasingly published in public health journals.
In regard to the methods, we have made it clear in the methods section of the paper that we used in-depth interviews and focus groups with the women participating in the intervention and their family members, and direct observation of the home visits of the field-level intervention deliverers. We would disagree to the comment that our data is “based solely on comments, experiences and anecdotes.” The methods we employed are used in qualitative research in health and nutrition to generate high quality in-depth and contextual data, which was required to generate the data for this process evaluation.
Comment: - In the results section, as regard to portions demonstrations, the authors report that women in the study confirmed that they learnt about how much rice and other foods they should eat in each meal. Did you evaluate the food consumption to assess if women really understood the meaning of right food portions?
Response: The team of the intervention trial has statistically evaluated the food consumption of the women which will be presented in a paper with the trial outcome results. Those statistical findings are beyond the scope of this current paper that presents qualitative process evaluation findings. To further clarify to the readers, we have now added the following text in the lines 79-81 of the revised manuscript.
“This paper presents the finding and insights obtained in the process evaluation. The impact assessments will be documented in a separate paper.”
We have further added the text below (lines 83-84) to further clarify the difference between the randomised controlled trail and the process evaluation study of this paper.
“To provide with the context of this process evaluation study, we will briefly describe the cluster randomised controlled trial (cRCT) within which we conducted the process evaluation.”
Comment: - In the results section, the authors report that women understood the importance of local products, or low-cost foods such as eggs and fish instead of meat, and that the families tried to increase the consumption of these foods. Was the consumption of these foods measured before the intervention and at the time of the interview? Is the increased/decreased consumption statistically and nutritionally significant?
Response: The design of the cRCT included baseline, end-line and cohort follow-up surveys. Analyses of the data from these surveys will be presented in the trial outcome paper. As a process evaluation manuscript, the current paper does not focus on measuring the dietary changes.
Comment: - Similarly, the authors report that some husbands said they changed their shooping patterns to accommodate healthier food. Did you check their food purchases before and after the nutritional intervention or the change is just self-declared by the subject?
Response: The before-after comparison of changes in practices will be described in the impact evaluation paper. We did not aim for such comparison in this process evaluation article.
Comment: - Is there a statistical analysis to support the results and conclusions? - Also, why were the women who followed the standard intervention not interviewed?
Response: I respond to these two comments combinedly. We as mentioned above, the statistical analysis is the aim of another paper that will present the impact outcomes of the randomised controlled trial. In this paper, we have presented the qualitative results that was analysed by using standard qualitative approaches, and our conclusions were drown on the basis of these qualitative data. We have not interviewed the women who belonged to the standard intervention (the control) because our aim in this paper was not to measure the impact of the intervention. However, as mentioned above, another paper reporting the impact outcome measures will make a case-control comparison of the changes in behaviours.
There are numerous examples in the published literature about articles written with qualitative process evaluation data that revealed participants’ own response to a health/nutrition intervention and the issues around adherence to the intervention. In this paper, we have the similar objective.
Comment: Although I generally agree that a demonstration intervention involving participants is more effective than a traditional passive lesson, in this case I find it difficult to accept the conclusion that "Programs to improve maternal diet quality should focus on promoting a healthy diet through practical demonstration of portion sizes and active engagement of the women and family instead of replicating the traditional information-based counselling" on the basis of the presented results.
Response: We agree to this comment. We have now omitted the sentence. The adjusted conclusion reads as below 426-432. We have deleted the sentence from the abstract too.
“We identified the strategies that worked to enhance adherence to the balanced plate nutrition education intervention in the context of the local setting. The findings provide an important message for future interventions for improving intervention uptake. Future implementation of the intervention should tailor the strategies to the local context to ensure optimal adherence to the intervention.”
Reviewer 2 Report
General Comment: The paper describes a study on intervention strategies for improving dietary consumption in pregnant women in Bangladesh. The strength of the study involves the practical application where participants become actively involved. The comments by these participants indicate successful intervention. This success is similar to that observed from active-learning exercises in the classroom compared to the traditional passive-learning from endless lectures. Moreover, the team-approach employed by the authors of involving relatives and neighbors in changing attitudes and behaviors related to nutrition during pregnancy appears to have made a significant difference. Overall, the paper is clearly written.
I address these concerns:
My major concern is the hypothesis as stated as stated beginning on line 83. This hypothesis is clearly stated and important to test. However, this was not tested in the study. Did the authors mean for this to be the actual hypothesis of the study or does it represent a future expectation? Based on the Results and Discussion, I assume the hypothesis of the study was that the intervention strategy as explained in the Methods would result in changing behaviors of pregnant women to consume a more nutritious diet that improves the health of the fetus, mother, and infant.
With regard to that hypothesis of infants of the intervention group being 100 grams heavier than the comparison group, is the average birth weight of infants of women in this geographical area known? Is that average birth weight lower than the healthy range? What is the birth weight range that is considered healthy?
The Results section is well written and the comments provided by the various participants support the overall conclusions made by the authors. While impressive, the Results are all narrative and anecdotal. It appears all data are from interviews and focus groups. Are there any available numbers or statistics from the coding methodology as described in the Data Analysis section?
Section 3.4 addresses potential financial aspects of converting to a healthier diet. Is there evidence that making these beneficial dietary adjustments will be more costly to the average family?
Line 153: Do the authors mean “compiled” instead of “complied”?
Author Response
Thank you for your constructive comment. Please find below our point-by-point response.
Comment: … This hypothesis is clearly stated and important to test. However, this was not tested in the study. Did the authors mean for this to be the actual hypothesis of the study or does it represent a future expectation? Based on the Results and Discussion, I assume the hypothesis of the study was that the intervention strategy as explained in the Methods would result in changing behaviors of pregnant women to consume a more nutritious diet that improves the health of the fetus, mother, and infant.
Response: We thank the reviewer for the comment. It gives us a chance to clarify the issue of hypothesis. The hypothesis we mentioned in the final subsection of the introduction was the hypothesis of the cluster randomised controlled trial (cRCT) for which we conducted this process evaluation study. This manuscript reports the findings of the process evaluation, not the results of the cRCT. We did not test any hypothesis in this process evaluation study as it is beyond the scope of a process evaluation. In the context of increasing demand of implementation assessment of community-based health and nutrition interventions to inform replicability and scaling up of programs, in this paper, we aim for exploring the key intervention strategies of the cRCT. For further clarity about the relationship between the cRCT and the study of this paper, we have now added the following text to the subsection in lines 83-84.
“To provide with the context of this process evaluation study, we will briefly describe the cluster randomised controlled trial (cRCT) within which we conducted the process evaluation.”
Comment: …is the average birth weight of infants of women in this geographical area known? Is that average birth weight lower than the healthy range? What is the birth weight range that is considered healthy?
Response: Yes, the average birth weight in the region is known. A study of 16,290 live born babies in the neighbouring two districts published by Klemm et al[1] in 2015 reported mean birthweight of 2433 grams. The World Health Organisation recommends 2500gm and more birthweight as healthy. Thus, the average birthweight in the study area is lower than the minimum healthy birthweight. In fact, the Klemm et al study found 55.3% of the neonates underweight and about 40% of them small for gestational age.
Comment: The Results section is well written and the comments provided by the various participants support the overall conclusions made by the authors. While impressive, the Results are all narrative and anecdotal. It appears all data are from interviews and focus groups. Are there any available numbers or statistics from the coding methodology as described in the Data Analysis section?
Response: In-depth interviewing and focus groups with different types of respondents including the intervention recipients and deliverers, and observation were the methods used in this study. The types of method were reflected in the nature of data that the reviewer has correctly noted. We did not quantify the findings, which we considered not very appropriate to qualitative analysis of this type of study. The statistical results, however, will be reported in another paper that focuses on the impact outcomes of the intervention.
Comment: Section 3.4 addresses potential financial aspects of converting to a healthier diet. Is there evidence that making these beneficial dietary adjustments will be more costly to the average family?
Response: Our findings revealed, in section 4.3, that the families endeavoured to manage the extra cost for advantageous food for pregnant women by adopting various strategies such as replacing usual food items with low-cost nutritious foods advised by the project counsellors, adjusting the family budget, and in rare occasions by working for some extra hours. While another paper that would present statistical analysis of the impact outcome indicators would assess the cost implication, our qualitative data suggests that the families were able to cope with the change in their food purchase.
Comment: Line 153: Do the authors mean “compiled” instead of “complied”?
Response: The type has been corrected.
[1]Klemm RDW et al. (2015). Low-birthweight rates higher among Bangladeshi neonates measured during active birth surveillance compared to national survey data. Maternal and Child Nutrition 11(4):583-594.
Reviewer 3 Report
Manuscript entitled „How can we improve the consumption of a nutritionally-balanced maternal diet in rural Bangladesh? The key elements of the ‘balanced plate’ intervention”
presents the findings of the process evaluation and describes the strategies that worked for intervention compliance to promote a balanced diet in pregnancy through practical demonstration in rural Bangladesh.
The manuscript consists of 1 table, and 47 references. In general manuscript is written well, clear, but it would be good to add table with „a balanced plate” – for readers’ information.
Author Response
Thank you for your useful comment. Please find below our point-by-point response.
Comment: … In general manuscript is written well, clear, but it would be good to add table with „a balanced plate” – for readers’ information.
Response: We thank the reviewer for the suggestion. We have described the balanced plate menu and added a table of the menu in the revised manuscript. The following text and the table have been inserted between lines 102 and 111.
“A culturally appropriate daily meal menu (Table 1) and a pictorial food chart were produced based on the currently recommended diet for Bangladeshi pregnant women. The menu contained foods from all seven-food groups (cereal, lentil, animal protein, vegetables, fruits, milk and oil) giving a daily 2500 kcal energy, and included essential micro and macronutrients. The balanced plate that we used for the demonstration was a combination of foods in appropriate portion size, essential to meet the requirements of a pregnant woman.
Round 2
Reviewer 1 Report
The authors have responded to my criticism and modified the manuscript.
I have no further comments.
Reviewer 2 Report
I am fully satisfied with all responses and revisions. I appreciate the clear explanations to my concerns.